# Functional Characterization of TetR-like Transcriptional Regulator PA3973 from *Pseudomonas aeruginosa*

**DOI:** 10.3390/ijms232314584

**Published:** 2022-11-23

**Authors:** Karolina Kotecka, Adam Kawalek, Magdalena Modrzejewska-Balcerek, Jan Gawor, Karolina Zuchniewicz, Robert Gromadka, Aneta Agnieszka Bartosik

**Affiliations:** Institute of Biochemistry and Biophysics, Polish Academy of Sciences, Pawinskiego 5a, 02-106 Warsaw, Poland

**Keywords:** *Pseudomonas aeruginosa*, TetR-like transcriptional regulator, PA3973, gene expression regulation, stress response

## Abstract

*Pseudomonas aeruginosa*, a human opportunistic pathogen, is a common cause of nosocomial infections. Its ability to survive under different conditions relies on a complex regulatory network engaging transcriptional regulators controlling metabolic pathways and capabilities to efficiently use the available resources. *P. aeruginosa PA3973* encodes an uncharacterized TetR family transcriptional regulator. In this study, we applied a transcriptome profiling (RNA-seq), genome-wide identification of binding sites using ChIP-seq, as well as the phenotype analyses to unravel the biological role of PA3973. Transcriptional profiling of *P. aeruginosa* PAO1161 overexpressing *PA3973* showed changes in the mRNA level of 648 genes. Concomitantly, ChIP-seq analysis identified more than 300 PA3973 binding sites in the *P. aeruginosa* genome. A 13 bp sequence motif was indicated as the binding site of PA3973. The PA3973 regulon encompasses the *PA3972-PA3971* genes encoding a probable acyl-CoA dehydrogenase and a thioesterase. In vitro analysis showed PA3973 binding to *PA3973*p. Accordingly, the lack of PA3973 triggered increased expression of *PA3972* and *PA3971*. The ∆*PA3972-71* PAO1161 strain demonstrated impaired growth in the presence of stress-inducing agents hydroxylamine or hydroxyurea, thus suggesting the role of *PA3972*-*71* in pathogen survival upon stress. Overall our results showed that TetR-type transcriptional regulator PA3973 has multiple binding sites in the *P. aeruginosa* genome and influences the expression of diverse genes, including *PA3972-PA3971*, encoding proteins with a proposed role in stress response.

## 1. Introduction

*Pseudomonas aeruginosa* is a bacterium commonly found in various ecological niches and characterized by the ability to survive in very unfavorable, frequently changing environmental conditions. It is also an opportunistic human pathogen, causing infections in immunocompromised and cystic fibrosis patients, and it is often isolated from infected pulmonary and urinary tracts, eyes, or burn wounds [1]. Currently, in the pandemic era of the Coronavirus disease (COVID-19) causing severe acute respiratory syndrome, *P. aeruginosa* is one of the most frequently detected co-existent bacterial pathogen with COVID-19 among hospitalized patients with 12% confirmed detections, along with *Haemophilus influenzae* (12%) and *Mycoplasma pneumoniae* (42%) [2]. *P. aeruginosa* can increase the risk of serious complications during infection and lead to higher morbidity and mortality. It can be easily transferred, because it can survive in varying conditions, such as aquatic environments, soil, plant, and animal tissues. The success of *P. aeruginosa* as a pathogen is largely connected to its intrinsic resistance to antimicrobial agents and the broad spectrum of acquired resistance mechanisms making the treatment of *P. aeruginosa* infections a growing problem [3]. In addition to virulence and resistance factors, these bacteria modulate metabolism in response to stress factors, including antibiotics [4,5]. Thus, it is essential to understand the mechanisms responsible for the adaptation of bacteria to various environmental conditions promoting survival and pathogenesis [6]. Gene expression control and cellular metabolism reprogramming play a pivotal role in adaptation. Elements responsible for the regulation of cellular processes, survival, and bacterial virulence during infection may be considered as potential new targets of anti-bacterial therapies [7].

*P. aeruginosa* is an example of a bacterium with an extensive regulatory network, in which transcriptional regulators comprise a crucial group controlling most cellular processes in response to environmental conditions [8,9]. A high number of genes encoding transcriptional regulators belonging to different families (492 genes in PAO1 reference strain) suggests their importance in *P. aeruginosa* cells [10,11]. The complexity of gene expression regulation in *P. aeruginosa* can be exemplified by the regulatory network controlling virulence or biofilm formation, which is tightly connected with the regulation of quorum sensing, antibiotic resistance, and pathogenicity [12,13].

The TetR/AcrR family of transcriptional regulators comprises an important group of regulatory factors that are well represented and widely distributed among bacteria [14]. They are involved in the control of various cellular processes, with known representatives engaged in the regulation of efflux pumps and transporters, conferring tolerance to toxic compounds and antibiotic resistance, like tetracycline repressor TetR from *Enterobacteriaceae* [15].

The TetR/AcrR-like regulators possess a two-domain structure with a DNA-binding helix-turn-helix (HTH) motif localized in the N-terminal part of the protein and a regulatory, ligand binding domain located at the C-terminus [15]. The C-terminal part contains several regions that can be involved in the binding of effectors (e.g., drugs) and in oligomerization, which can also serve as a signal to modulate the activity of the regulator. Most of the TetR-like regulators act as dimers (e.g., TetR and CamR), while others (e.g., QacR) bind a cognate operator as a pair of dimers; EthR seems to bind as an octamer [16,17,18,19]. The crystal structures of TetR family members showed conserved structures built of 9–12 helices with the DNA-binding domains formed by three helix bundles and part of the adjacent helix [18]. The second and third helices create an HTH motif, with the third called the recognition helix, as it inserts into the DNA major groove [14].

The archetype of this family of bacterial transcriptional regulators, TetR encodes a part of the most common resistance mechanism against tetracycline in Gram-negative bacteria [14]. The regulator binds as a dimer to two palindromic operator sites *tetO* repressing the expression of *tetA*, a gene encoding the tetracycline exporting protein [18]. In the presence of tetracycline, the complex of TetR with antibiotics and a magnesium ion is created, abolishing TetR binding to DNA and leading to derepression of the *tetA* and *tetR* genes [20]. The other example of regulation of the drug efflux operon by a TetR family member is the AcrR repressor involved in the regulation of drug efflux operon *acrAB* in *Escherichia coli* [15,21]. Products of this operon are part of the AcrAB-TolC tripartite transporter that effluxes substrates from the periplasm across the outer membrane out of the cell [22] and concomitantly a wide range of compounds may modulate AcrR binding to DNA [21]. Studies involving multiple TetR-type regulators indicated that these proteins may act as sensors of the environment to regulate the expression of genes in response to various stimuli [14,15]. Several of these factors are engaged in the repression of genes for cell envelope permeability and membrane transport. They regulate the expression of genes involved in antibiotic production, stress response, modulation of basic metabolism, transport systems, resistance to antibiotics, and host-encoded defense mechanisms, as well as pathogenesis [23].

The 38 representatives of the TetR family are encoded in the *P. aeruginosa* PAO1 reference genome, but only a few were characterized in detail. The best known is MexZ (PA2020), which represses the *mexXY* operon encoding the multidrug transporter, which confers resistance to various antimicrobials, including aminoglycosides [24]. The MexZ dimer binds to a 20-bp palindromic sequence in the intergenic region between *mexXY* and *mexZ* genes repressing their transcription [25,26]. The repression is alleviated in response to aminoglycosides, tetracyclines, and macrolides or oxidative stress [27,28,29,30], which does not involve MexZ interaction with antimicrobials but depends on the action of MexZ anti-repressor ArmZ (PA5471) [26,31,32], which targets the N-terminal domain of MexZ [33].

Another characterized member of the family, CifR (PA2931), acts as a repressor of the *cif* (*PA2934*) gene encoding a virulence factor secreted by *P. aeruginosa* known as CFTR inhibitory factor [34,35]. Whereas most members of the TetR family act as repressors, some, including *P. aeruginosa* Pip (PA0243), were also shown to be involved in the positive regulation of gene expression, in the case of Pip, genes engaged in production of pyocyanin, another potent virulence factor [36]. 

The PA3973 protein from *P. aeruginosa* has been classified in silico as a putative TetR-type transcriptional regulator. Based on the PseudomonasNet, a genome-wide functional network of *P. aeruginosa* genes, a negative regulation of transcription, DNA-dependent, and cellular response to stress are predicted for PA3973 according to GO terms [37]. The *PA3973* gene was previously identified as significantly upregulated in cells lacking *vqsM*, a crucial component of quorum sensing [38], but also in *P. aeruginosa parA* and *parB* mutants, which showed disturbed chromosome segregation [8,39]. This study aimed to decipher the function of this regulator.

## 2. Results

### 2.1. Overview of the PA3973 from P. aeruginosa

PA3973 is the first gene of the *PA3973-PA3970* gene cluster in the *P. aeruginosa* PAO1 genome, encoding a protein classified as a TetR-type putative transcriptional regulator (Figure 1A). The structure prediction showed two potential domains in PA3973 with the N-terminally located HTH motif encompassing amino acids 14–74 (helices α2 and α3) possibly involved in contact with DNA [40] and the C-terminal part, creating a putative ligand-binding domain responsible for signal perception and dimerization (Figure 1B). Two monomers of PA3973 possibly form a dimer (Figure 1C), similarly to other members of the TetR family [14,15]. The ability of the protein to self-assemble was confirmed using glutaraldehyde crosslinking of purified His_6_-PA3973 followed by a Western blot analysis (Figure 1D). The assay showed additional bands migrating just above the target dimer, which likely reflects various conformational states of the crosslinked dimer.

In silico analyses and database mining suggested that products of the genes encoded within the *PA3973-PA3970* gene cluster could be involved in a stress response (Figure 1A). They encode a probable acyl-CoA dehydrogenase, *E. coli* AidB homolog (PA3972), a putative thioesterase from PaaI family (PA3971), and an AMP nucleosidase, Amn (PA3970). According to the Pseudomonas Database, the *PA3973-PA3970* genes are predicted to form an operon, but since experimental evidences are lacking, throughout this manuscript, the term gene cluster is used.

### 2.2. Identification of PA3973-Regulated Genes and Binding Sites for This Transcriptional Regulator in the P. aeruginosa Genome

To analyze the role of PA3973 in *P. aeruginosa*, a PAO1161 ∆*PA3973* mutant was constructed. This mutant strain did not display obvious differences in growth in LB or M9 medium, swimming, swarming, twitching, or biofilm formation compared to the wild type (WT) parental strain PAO1161 (Figure 2A,B). In parallel, the effect of *PA3973* overexpression was tested by linking the gene to an IPTG (isopropy-β-D-thiogalactopyranoside)-inducible promoter in plasmid pKKB3.11 (*lacI^q^*-*tac*p-*PA3973*). No effect of PA3973 overproduction on bacterial growth was observed in *P. aeruginosa* under the tested gradient of IPTG in the growth medium (Figure 2C).

To identify genes that display PA3973-dependent expression, we used RNA sequencing analysis (RNA-seq). In addition, we performed chromatin immunoprecipitation and sequencing analysis (ChIP-seq) to identify PA3973 binding sites in the *P. aeruginosa* genome. The rationale behind an analysis of cells with PA3973 in excess was based on the following: (i) the relatively low level of *PA3973* expression under standard growth conditions (LB or M9 medium; Bartosik AA, personal communication); (ii) the likelihood that an excess of *PA3973* might mimic the induced, activated state of the protein; and (iii) the fact that the effector, signal, or partner for this TetR-type regulator is unknown.

RNA-seq was performed using material isolated from cultures of the strains PAO1161 pKKB3.11 (*tac*p-*PA3973*, hereafter called PA3973+) and PAO1161 pAMB9.37 (*tac*p, empty vector [EV]) grown in a selective LB medium supplemented with 0.05 mM IPTG (Figure 2C; Appendix A). The comparison of the PA3973+ and EV transcriptomes showed 648 loci with altered expression (fold change [FC] ≤ −2 or ≥2, adjusted *p*-value ≤ 0.01) (Figure 3A; Appendix A). The expression of 374 loci was downregulated, whereas 274 loci displayed increased expression. For convenience, we use the *P. aeruginosa* PAO1 gene names throughout the manuscript, although the corresponding PAO1161 gene names are also included. The functional classification of the identified loci, based on PseudoCAP [11], showed that the upregulated genes were mostly associated with protein secretion/export systems, non-coding RNA, and cell wall functions (Figure 3B; Appendix A). Decreased expression was observed for several genes encoding proteins engaged in energy metabolism, chemotaxis, central intermediary metabolism, or adaptation and protection. The most significantly downregulated genes were *norC* from the *PA0509-PA0527* gene cluster, *nosR* from the *PA3391-PA3393* cluster, *narK1* from the *PA3874-PA3880* gene cluster, *ccoP2* from the *PA1555-PA1557* cluster, *arcD* from the *PA5170-PA5173* operon, or *nrdD* from the *PA1919-PA1920* gene cluster. Concomitantly, the most significantly upregulated loci were *PA5024*, *PA4193* from the *PA4191-PA4195* cluster, *cysT* from the *PA0280-PA0284* cluster, *PA3445-PA3446*, *PA2311*, and *PA3530* (Figure 3B; Appendix A). 

The RT-qPCR (reverse transcription followed by quantitative PCR) analysis was performed to validate the changes in gene expression in the PA3973 overproducer strain in an independent set of biological replicates, collected at the same conditions as used for the RNA-seq analysis (Figure 3C). The genes selected for the RT-qPCR analysis had a broad range of fold changes observed in PA3973+ vs. EV cells, and the selected reference gene *proC* (*PA0393*) was not altered in the tested conditions based on RNA-seq results. Essentially, most of the RT-qPCR results correlated well with the RNA-seq data, confirming a negative impact of PA3973 on the expression of *PA3614*, *PA5208*, *PA5460*, *PA5497*, and *PA3972* (Figure 3C).

To identify PA3973 binding sites in the *P. aeruginosa* genome, ChIP-seq analysis was performed using anti-FLAG antibodies and ∆*PA3973* cells carrying plasmid pMEB255 (*tac*p-*PA3973-flag*), which were grown in a selective LB medium supplemented with 0.05 mM IPTG. As a background control for the ChIP procedure, the ∆*PA3973* strain carrying plasmid pABB28.1 (*tac*p-*flag*) was grown under the same conditions, and samples were processed in parallel. The comparison of PA3973-FLAG ChIP samples with control samples using a fold enrichment (FE) cut-off value of 2 (Figure 4A) yielded 308 PA3973-FLAG ChIP-seq peaks. The summits of 139 peaks (45%) mapped to intergenic regions (Appendix A), suggesting the potential of PA3973 to regulate the expression of adjacent loci. The summits of 169 peaks mapped to gene bodies (coding regions) (Appendix A). A search for nucleotide motifs shared by sites bound by PA3973 using MEME [42] showed the presence of a 13 bp motif SAAGRNMTGAACG, where S-G or C; R-A or G; M-A or C (Figure 4B) based on the analysis of 100 bp regions encompassing summits of all 308 peaks indicating PA3973 binding sites.

The functional PseudoCAP analysis of genes potentially influenced by PA3973, based on the ChIP-seq results, demonstrated enrichment in genes encoding proteins engaged in secretion/export systems, adaptation and protection, and central intermediary metabolism (Figure 4C). Table A1 (Appendix B) lists the most enriched PA3973 bound sites (FE > 4) under the tested conditions and adjacent genes, in which expression could be modulated. This data indicates that PA3973 has multiple binding sites in the *P. aeruginosa* genome, which suggests that this factor may function as a crucial component of the gene expression regulatory network.

### 2.3. Genes under the Direct Control of PA3973

Importantly, 38 of the 648 genes showing altered expression in response to a *PA3973* excess possessing a binding site for this transcriptional regulator within their promoter regions or adjacent to these genes (Figure 4A and Table A2). In addition, 26 detected in coding regions were located in the vicinity of genes that showed changes in expression level in RNA-seq analysis (Appendix A), but the mechanism by which PA3973 could influence their expression requires further studies.

The ChIP-seq analysis indicated that PA3973 binds within the region preceding its own coding sequence, in agreement with its effect on the expression of the downstream located *PA3972-PA3971* genes (Table A1). The PA3973 binding site with the second highest fold enrichment, besides the *PA3973* promoter region, was detected in the putative promoter of *PA4156*, encoding a protein highly homologous to the *Vibrio cholerae* vibriobactin receptor (ViuA) involved in iron acquisition [43]; however, no significant changes in *PA4156* expression were detected under tested conditions in our RNA-seq analyses (Table A1).

A PA3973 binding site was also detected in the putative promoter of *PA1673*, the gene showing the most severe downregulation in the RNA-seq analysis (Table A2), as well as in promoter regions of *PA3762* and *PA3531* that were downregulated in response to PA3973 excess (Table A2). Interestingly, the *PA3530*, which was transcribed up-stream from the *PA3531* gene, also showed changes in expression under PA3973 overproduction but with significantly increased mRNA level (Table A2). 

Similarly, the *PA2468*, encoding a sigma factor FoxI [44], had a peak in the region preceding the gene (Table A2). Interestingly, three other genes coding for RNA polymerase sigma factors *PA2426*, *PA3899*, and *PA0715* exhibited changes in expression in response to PA3973 and possess PA3973 binding sites in their promoter regions (Table A2).

To confirm the interactions of PA3973 with the selected regions, we performed electrophoretic mobility shift assays (EMSA) using purified PA3973-His_6_ and DNA fragments corresponding to the putative promoter regions of the genes exhibiting the highest fold enrichment in ChIP-seq analysis: *PA3973*, *PA4156*, *PA4710*, *PA0061*, *PA2722*, *PA0195,* and *PA2468* (Table A1). PA3973-His_6_ was able to specifically bind to *PA3973*p in a concentration-dependent manner (Figure 5A). The EMSA assay also showed PA3973-His_6_ interactions with putative promoters of *PA0061*, *PA0195*, and *PA2722* (Figure 5C,D,F). No clear shifted DNA–protein complexes, but a disappearance of unbound specific DNA in the presence of the higher concentration of the protein, could be observed in the case of the *PA2468*p, *PA4156*p, or *PA4710*p (Figure 5E,G,H) in tested conditions. It is possible that additional factors (e.g., a cellular partner, and/or the presence of the ligand) might be required for the efficient PA3973 binding to these sites.

All tested promoters contained sequences matching the motif SAAGRNMTGAACG predicted using MEME (Figure 4B and Figure 5I). In the case of *PA3973*p, this motif appeared twice and is a part of palindromic and pseudopalindromic sequences (TGAATCC GGATTCA and TGAAGCG TGATTCA, respectively) encompassing the predicted −10 sequence of *PA3973*p (Figure 5I). To verify the importance of the identified palindromic sequence in DNA binding by PA3973, a variant of the *PA3973* promoter fragment (*PA3973*p*) lacking the fragment encompassing the TGAATCCGGATTCA palindrome and a part of the pseudopalindrome was tested in an EMSA. PA3973-His_6_ was not able to bind to this variant of *PA3973*p, suggesting the involvement of these sequences in the protein recruitment to this promoter (Figure 5B,I).

To further examine the influence of PA3973 on the expression of the aforementioned genes, their promoter regions were cloned upstream of a promoter-less *xylE* gene in the vector pPTOI. Out of the selected genes, only *PA2468* and *PA4156* promoters were active in the heterologous host *E. coli* DH5α (Figure 5J). The expression of *PA3973* in cells carrying a plasmid with *PA4156*p-*xylE* resulted in significantly reduced XylE activity in the corresponding cell extracts, but no such regulation was observed for *PA2468*p-*xylE* under the tested conditions (Figure 5J).

To further define the targets of PA3973 action, we performed an RNA-seq analysis of PA3973-deficient cells relative to WT cells, which showed significantly increased *PA3972* and *PA3971* transcript levels, confirming the role of PA3973 in *PA3972-PA3971* repression (Figure 5K; Appendix A). Additionally, three genes, *PA2174*, *PA5497*, and *D3C65_10195*, also exhibited increased mRNA levels in the absence of PA3973, indicating a possible involvement of this regulator in their control. The RT-qPCR verification of the changes observed in the RNA-seq analysis showed the direction of changes to be consistent for both analyses, confirming the influence of PA3973 on the expression of these genes (Figure 5K). Overall, this data confirmed that PA3973 binds to DNA fragments identified in the ChIP-seq analysis and may regulate the activity of target promoters to influence gene expression.

### 2.4. Towards the Biological Function of PA3973-PA3971 Gene Cluster

In silico analyses and database mining suggested that the products of the *PA3973–PA3971* gene cluster could potentially be involved in stress response. Besides the first gene coding for the TetR-type transcriptional regulator PA3973, this cluster also encodes a probable acyl-CoA dehydrogenase (PA3972) and a putative thioesterase (PA3971). The PA3972 from *P. aeruginosa* is the putative homolog of the AidB protein from *E. coli* (41% identity, 56% similarity) with possible acylo-CoA dehydrogenase activity and oxidoreductase activity, acting on the CH-CH group of donors [11]. The AidB protein is involved in the defense against methylating agents [45,46,47]. In *E. coli*, *aidB* was upregulated in response to small doses of DNA-methylating agents [48]. This gene, along with *alkA* and *alkB*, belongs to the Ada protein regulon that controls the cellular stress response induced by the presence of alkylating agents, leading to changes in DNA or RNA [49]. 

The *PA3971* gene encodes a hypothetical protein belonging to the thioesterases family [11]. Its expression level increases when the strain is subjected to stress factors such as hydrogen peroxide [50]. A BLAST analysis showed approximately 40% similarity to a thioesterase from *E. coli*, which is responsible for the degradation of phenylacetic acid [51], and a high 91% similarity to a putative esterase from *Acinetobacter baumanii* with a role in the biosynthesis of secondary metabolites.

To assign a biological function to *PA3972* or *PA3971*, the chromosomal double mutant of PAO1161 in *PA3972-71* genes was constructed. No changes in colony morphology, bacterial motility, or biofilm formation were observed between the ∆*PA3972-71* mutant and the WT strain. Lack of *PA3973* or *PA3972-71* did not significantly affect the growth of the cells in a rich medium, or minimal M9 or MOPS medium supplemented with glucose as the carbon source in the early stages of culture growth (Figure 6A,D,G). However, prolonged incubation over eight hours (and up to 48 h in the case of a MOPS medium) resulted in a significantly lower OD_600_ of the ∆*PA3972-71* mutant and to a less extent for ∆*PA3973* in comparison with the WT strain (Figure 6G). This suggests that in conditions that might involve stress possibly connected with starvation, the PA3973, PA3972, and/or PA3971 proteins might play an important role in facilitating pathogen survival. The growth in the presence of hydroxylamine and hydroxyurea, two stress-inducing agents, was also tested (Figure 6B,C,E,F). Hydroxylamine exposure introduces mutations by acting as a DNA nucleobase amine-hydroxylating agent [52], whereas hydroxyurea is a ribonucleotide reductase inhibitor, altering DNA replication and nucleotide metabolism by the depletion of deoxyribonucleoside triphosphate pool [53].

The presence of hydroxylamine (50 µg/mL) or hydroxyurea (1–20 mM) tested in an LB and M9 minimal medium with glucose slowed down the growth of *P. aeruginosa* strains in comparison to growth in a medium without these compounds (Figure 6A–F). The presence of hydroxylamine caused a prolonged lag time of bacterial culture growth with a significantly slower kinetic of the ∆*PA3972-71* strain in comparison with the WT or ∆*PA3973* in M9 medium (Figure 6E), indicating that PA3972 and/or PA3971 could facilitate the growth of bacteria in such conditions. 

The significantly slower growth of ∆*PA3972-71* cultures in comparison with the WT, was also observed in the presence of 20 mM (in LB medium) or 1 mM (in M9 medium) hydroxyurea (Figure 6C,F). This data indicates that PA3972 and/or PA3971 may play a role in stress response; however, further studies are needed to discover the underlying mechanisms.

The RT-qPCR analysis was also applied to check the expression of *PA3973* and selected genes from its regulon in the *P. aeruginosa* WT cells exposed to antibiotics and hydroxylamine. The analysis showed no significant difference in the *PA3973* expression upon the addition of carbenicillin compared to control cultures (non-treated grown in LB), but significantly decreased the expression in response to kanamycin, and increased the mRNA level of *PA3973* in response to tetracycline (Figure 6H). Interestingly, the expression of *PA3972-PA3970* genes was significantly downregulated in response to tetracycline (Figure 6H), which correlates with the increased expression of *PA3973*, confirming its repressor functions for the *PA3972-71* genes. Moreover, in the presence of hydroxylamine, the expression of *PA3973* was significantly down-regulated, and the *PA3972*, but not *PA3971* or *PA3970*, showed a notable up-regulation, which suggests that *PA3972* expression is induced in response to hydroxylamine. 

This data indicates that PA3973 action might be connected with the cellular response to some stressors, but further studies are needed to discover the underlying mechanisms and the role of PA3973-dependent genes, including *PA3972,* in this process.

## 3. Discussion

The PA3973 of *P. aeruginosa* based on sequence similarity is classified as the TetR-type transcriptional regulator. The N-terminal part, similar to other members of the TetR family, consists of the HTH DNA-binding domain predicted with structure modeling, in which helices α2 and α3 are involved in direct contact with cognate DNA [14,15]. The C-terminal part is presumably involved in signal perception and oligomerization. The presence of a dimeric form was confirmed with glutaraldehyde cross-linking experiments with the use of purified His_6_-PA3973. The signal to which PA3973 responds in *P. aeruginosa* awaits elucidation.

The ChIP-seq was applied to evaluate the binding of PA3973 across the chromosome in exponentially growing *P. aeruginosa* cells. The 308 PA3973-binding targets located across the genome, including intergenic and coding regions, were identified. Multiple binding sites in the genome suggest the role of PA3973 as a global regulator of gene expression in *P. aeruginosa*. However, the transcriptional profiling of ∆*PA3973* mutant cells grown in rich medium did not support the global impact. Significantly, under this conditions, the increased expression of *PA3972*, *PA3971*, *PA2174*, and *PA5497* genes was detected in PA3973 deficient cells in comparison with the WT (Figure 5K; Appendix A). In contrast, the RNA-seq analyses of exponentially growing cells overproducing PA3973 showed 648 genes responding to PA3973 (Appendix A). Among them 38 genes possess PA3973 binding sites in their promoter regions or in the vicinity of these genes and the additional 26 genes in coding regions (Table A2, Appendix A). These genes could exhibit direct PA3973-dependent regulation in contrast to the remaining loci with altered expression in *PA3973* overexpressing cells, which likely represent the indirect effect of PA3973 action on *P. aeruginosa* cells.

In the case of most transcriptional regulators their high-affinity binding sites are largely restricted to non-coding DNA regions to exert direct effect on the target genes’ expression [54]. In the case of PA3973, more than half (169 out of 308) of the DNA targets for PA3973 are located within the coding regions of the genome and appear to have little or no effect on gene expression, at least under the conditions tested. This may suggest that additional factors (e.g., special growth conditions, presence of ligand, signal and/or cellular partner) might be involved in PA3973 regulatory activity. A similar pattern was observed; for example, in *E. coli,* RutR, the uracil-responsive transcription factor from the TetR family involved in the regulation of the expression of the *rut* operon encoding genes for the catabolism of pyrimidine. RutR showed multiple additional binding sites within genes; however, their role in gene expression control remains elusive [55]. Concomitantly, an interesting mode of action among TetR-like regulators exemplifies a LuxR protein from *Vibrio alginolyticus* [56]. It controls the expression of approximately 280 genes that contain either a symmetric palindrome or asymmetric binding motifs upstream, overlapping start codon, or inside open reading frames. Differences in LuxR binding to these motifs, modulated by its N-terminal extension, allow the repression or activation of cognate gene expression to impact quorum sensing and virulence [56]. Based on ChIP-seq data and the localization of identified PA3973 binding sites, we speculate that this regulator might present a similar mode of action.

The sequence analysis of PA3973 ChIP-seq peaks revealed a 13-bp binding motif with the consensus SAAGRNMTGAACG. In the case of *PA3973*p, this motif was found twice in the sequence encompassing the predicted −10 promoter sequence (Figure 5I). Two closely located motifs, separated by a dozen or more base pairs, are also present in other loci identified as PA3973 targets (e.g., *PA0061*p, *PA0195*p, *PA2468*p, and *PA4710*p). One PA3973-binding motif was identified in *PA2722*p, *PA4056*p, or *PA4156*p. In some target regions, the binding sites for other regulatory factors could be detected, such as the Fur (ferric uptake regulation protein) binding site in *PA4156*p or *PA4710*p. This suggests that, besides PA3973, other factors might be involved in the regulation of these genes. The Fur is involved in the regulation of iron metabolism in *P. aeruginosa* (e.g., in the control of the production of receptors and siderophores involved in iron scavenging) [57]. Moreover, the PA3973 binding site was identified in the promoter region of the *PA2468* encoding extracytoplasmic function sigma factor (ECF) FoxI [44].

Other interesting regulatory connections are the decreased expression of *PA1920* in response to PA3973 excess (Table A1 and Appendix A) and the up-regulation of *PA5497* in Δ*PA3973* mutant (Figure 5K; Appendix A). The *PA1920* and *PA5497* encode a class III (anaerobic) ribonucleoside-triphosphate reductase subunit, NrdD, and a class II (cobalamin-dependent) ribonucleotide-diphosphate reductase subunit, NrdJa, respectively. Importantly, it was shown that both ribonucleotide reductases (RNR) are highly expressed during infection [58]. In *P. aeruginosa*, three different types of RNR are encoded, allowing adaptation to different environmental conditions including anaerobiosis [59,60]. The expression of RNR genes is under the control of NrdR, a global regulator, which, in *E. coli*, binds to the NrdR boxes (acaCwAtATaTaTwGtg) in the promoters of target genes [60]. Such sequences could be found in *PA1920* and *PA5497* promoter sequences, and the ChIP-seq data showed the PA3973 binding site in the coding sequence of *PA1920* and *PA5497*. Moreover, the PA3973 binding site with fold enrichment 10 (Table A1) was detected in the predicted *PA4056-PA4057* (*nrdR-ribD*) operon, overlapping the end of the *nrdR* and the beginning of the *ribD* ORFs. It suggests the existence of complex regulatory interdependencies, managing the expression of these genes.

The EMSA assays for the selected targets of PA3973 showed varying affinity of PA3973 to analyzed targets, with the strongest binding observed in the case of *PA3973*p comprising two pseudopalindromic/palindromic sequences, which likely served as the operator sequences recognized by PA3973 (Figure 5A–H). In the promoter region of PA3973, the identified PA3973 binding sites are part of two closely spaced pseudopalindromic/palindromic sequences of 14 bp in length separated by 11 bp (Figure 5I). Their arrangement and localization resemble the operator sequences of other TetR-type regulators (e.g., TetR with two 15 bp operator sequences (TCTATCATTGATAGG) separated by 11 bp) [14,18]. The deletion of the palindromic sequence and part of the psedopalindromic sequence prevented PA3973 binding to truncated *PA3973*p*, suggesting that the presence of these sequences is needed for protein binding to cognate sequence and to exert a possible effect on expression (e.g., repression). It is tempting to propose the model in which PA3973 acting as a dimer binds to these operator sequences, repressing expression from its own promoter and possibly *PA3972-PA3971* genes.

Our studies on *P. aeruginosa* growth in the presence of stress-inducing agents like hydroxylamine or hydroxyurea showed that PA3972 could facilitate bacterial growth in such detrimental conditions. Interestingly, it was recently shown that PA3972 belongs to negative regulators of bacterial virulence as predicted with PseudomonasNet and further validated by the analysis of the *PA3972* knockout mutant in the *Caenorabditis elegans* infection model [37]. The disruption of *PA3972* resulted in elevated *P. aeruginosa* PAO1 virulence in *C. elegans*.

Taken together, these results demonstrated the participation of TetR-type transcriptional regulator PA3973 in several diverse cellular processes including the modulation of cellular response to growth conditions to keep cellular homeostasis and facilitate pathogen survival. Some of the identified PA3973 targets play a role in *P. aeruginosa* stress response and pathogenesis, but the exact mechanism and the signals to which PA3973 responds and regulates its targets need further investigation. Presented data brings us closer to understanding the complex *P. aeruginosa* system, to draw the picture of regulatory connections involved in controlling bacteria lifestyle as a free-living organism and during infection.

## 4. Materials and Methods

### 4.1. Bacterial Strains, Plasmids, and Growth Experiments

The bacterial strains used and constructed in this study (listed in Appendix A) were grown in an L broth (LB) rich medium or on LB-agar at 37 °C, or in M9 or MOPS minimal medium [61] supplemented with glucose (0.5%) as the carbon source, and with leucine (10 mM) added in the case of *P. aeruginosa* PAO1161 *leu*^−^ strains cultured in minimal media. Hydroxyurea (1 or 20 mM) and hydroxylamine (50 µg/mL) were used in growth experiments as indicated. For the selection of plasmids in *E. coli*, the media were supplemented with 10 µg/mL chloramphenicol, 50 µg/mL kanamycin, or benzyl penicillin at a final concentration of 150 µg/mL in a liquid medium or 300 µg/mL in agar plates. For *P. aeruginosa* strains, carbenicillin (300 µg/mL), rifampicin (300 µg/mL), and chloramphenicol (75 µg/mL in a liquid medium; 150 µg/mL in plates) were used as required. For growth experiments, cultures were grown overnight with shaking at 37 °C and diluted with fresh medium either in flasks, closed with cotton plugs, or in 96-well plates. Bacterial growth was monitored by measurements of optical density at 600 nm (OD_600_) in a spectrophotometer Shimadzu UV-1800 (Shimadzu Corporation, Kyoto, Japan) or with the use of Varioskan Lux Multimode Microplate Reader and SkanIt RE software (Thermo Fisher Scientific, Waltham, MA, USA). Competent *E. coli* cells were prepared with a treatment of CaCl_2_ [62]. The competent *P. aeruginosa* cells were prepared with a treatment of MgCl_2_ as described previously [63]. 

All plasmids used and constructed in this study are described in Appendix A. All primers used are listed in Appendix A. 

The *P. aeruginosa* PAO1161 Δ*PA3973* and Δ*PA3972-71* mutants were obtained with allelic exchange. The competent cells of *E. coli* S17-1 were transformed with plasmid pKKB60.6 or pSOB3 (derivatives of suicide vector pAKE600) to create the donor strain, and the WT *P. aeruginosa* PAO1161 Rif^R^ was used as the recipient. The allele exchange procedure was performed as described previously [64]. The verification of the allele exchange was performed with PCR.

### 4.2. Motility and Biofilm Formation Assays

Motility assays were performed as described previously using cultures in an LB medium [65]. To standardize the assay, all plates contained the same volume of the medium. The biofilm amount was measured with the crystal violet staining method [66].

### 4.3. Protein Purification

The *E. coli* BL21(DE3) strains transformed with pKKB28.3 or pMEB265 encoding a His_6_-PA3973 or PA3973-His_6_ fusion, respectively, were grown to exponential phase in an LB medium with the supplementation of 0.5 mM IPTG. The cells were harvested with centrifugation, resuspended in a LEW buffer [50 mM sodium phosphate buffer, pH 8.0; 300 mM NaCl] supplemented with lysozyme (1 mg/mL), PMSF (1 mM), and benzonase nuclease (250 U, Sigma), and sonicated. His_6_-tagged proteins were purified from the cell lysate using chromatography on Ni-agarose columns (Protino Ni-TED 1000, Macherey-Nagel, Düren, Germany) with 300 mM imidazole in LEW buffer used for elution. The purification procedure was monitored by SDS-PAGE using a Pharmacia PHAST gel system. The fractions containing the purified protein were dialyzed overnight in 50 mM Tris-HCl buffer (pH 8.0) containing 5% (*v*/*v*) glycerol and stored in aliquots at −80 °C.

### 4.4. Cross-Linking of Purified Protein

The purified His_6_-PA3973 was cross-linked with increasing concentrations of glutaraldehyde as previously described [67]. Samples were then suspended in the loading buffer [50 mM Tris-HCl (pH 8.0), 0.1 M DTT, 2% SDS, 0.1% bromophenol blue, 10% glycerol], boiled, and separated on an SDS-PAGE gel, transferred to nitrocellulose membrane (Amersham Protran, Cytiva, Germany), and used in a Western blot analysis with the use of anti-His_6_ antibodies.

### 4.5. RNA Isolation, RNA-Seq

The total RNA was isolated from three independent replicate samples of *P. aeruginosa* PAO1161 overexpressing the *PA3973* gene, samples of control strain carrying the empty vector, as well as *P. aeruginosa* PAO1161 WT and Δ*PA3973* strain. RNA isolation, sequencing, as well as data analysis were performed essentially as described previously [68]. The raw data is available in the NCBI‘s Gene Expression Omnibus (GEO) database (http://www.ncbi.nlm.nih.gov/geo/) under accession number GSE211771 (released 22 November 2022).

### 4.6. RT-qPCR Analyses

For the qRT-PCR analyses, cells from the PAO1161 pKKB3.11 (*lacI*^q^-*tac*p-*PA3973*) and PAO1161 pAMB9.37 (*lacI*^q^-*tac*p) strains grown in a selective LB medium supplemented with 0.05 mM IPTG, or from PAO1161 WT and Δ*PA3973* mutant cultures grown in the rich medium were collected from 2 mL of cultures at an optical density at 600 nm 0.5. For the testing influence of stress conditions on the expression of selected genes, the RNA was isolated from the WT PAO1161 cultures grown in a rich medium supplemented with subinhibitory concentrations of carbenicillin (32 µg/mL), kanamycin (10 µg/mL), tetracycline (4 µg/mL), or hydroxylamine at concentration 50 µg/mL. The RNA was isolated from 2 mL of cultures collected at an optical density at 600 nm 0.5.

Three biological replicates of total RNA (2 µg per reaction) from each strain served as a template for cDNA synthesis with TranScriba Kit (A&A Biotechnology, Gdansk, Poland). The cDNA was used as a template in qPCR performed with 5xHOT FIREPol EvaGreen qPCR Mix Plus (Solis Biodyne, Tartu, Estonia). Three technical replicates per each gene/sample were used. The sequences of primers used for RT-qPCR analysis are listed in Appendix A. The efficiency of the quantitative PCR reaction with each primer pair was calculated and used to calculate the ratio of each studied gene to the reference gene. All primer pairs used showed the efficiency of amplification between 0.95–1.05. Changes in individual gene expression between the mutant and WT strain, between PA3973 overproducer and control strain, or between treated and untreated cells were calculated with the normalization of Cp values to mean Cp value for a *proC* (*PA0393*) reference gene using the Pfaffl method [69]. The qPCR was performed using the Light Cycler 480 (Roche). PCR products were detected with SYBR green fluorescent dye and amplified according to the following protocol: one cycle at 95 °C for 15 min followed by 40 cycles at 95 °C for 15 s and 60 °C for 20 s. Each 18 µL reaction contained 3.6 µL 5× reaction mix, 1 µL of five times diluted cDNA, and 1.5 µL of mixed 6 µM primers. In each run, negative controls (no cDNA) for each primer set were included.

### 4.7. Chromatin Immunoprecipitation and Sequencing

The ChIP-seq was performed as previously [68]. *P. aeruginosa* Δ*PA3973* strain carrying pMEB255 (*tac*p-*PA3973*-*flag)* or pABB28.1 (*tac*p-*flag)* vectors were grown in rich medium containing chloramphenicol (75 µg/mL) and 0.05 mM IPTG as the inducer. The ChIP procedure was performed using anti-FLAG antibodies (MA1-91878, Invitrogen). Data were processed essentially as previously [68]. The sequencing data is available in the NCBI‘s Gene Expression Omnibus (GEO) database (http://www.ncbi.nlm.nih.gov/geo/) under accession number GSE211769 (released 22 November 2022).

### 4.8. In Vitro Protein-DNA Interactions

The electrophoretic mobility shift assay with purified PCR products (~100 ng in reaction) and PA3973-His_6_ was conducted in a binding buffer [10 mM Tris-HCl (pH 8.5), 10 mM MgCl_2_, 100 mM KCl, 0.1 mg/mL BSA]. The tested DNA fragments were amplified using the appropriate pair of primers listed in Appendix A: #13/#4 for *PA3973*p; #13/#14 for *PA3973*p*; #15/#16 for *PA0061*p; #17/#18 for *PA0195*p; #19/#20 for *PA2468*p; #21/#22 for *PA2722*p; #23/#24 for *PA4156*p; #25/#26 for *PA4710*p. Approximately 300 ng of competitor DNA fragment, obtained by the annealing of oligonucleotides #27/#28 (Appendix A), was used as an internal control in each reaction. The DNA fragments were incubated in a binding buffer with the increasing concentration of purified PA3973-His_6_ at 23 °C for 30 min. All reactions were prepared in a total volume of 20 μL. The complexes were separated on a 1.5% agarose gel in 0.5× Tris-borate-EDTA (TBE) buffer at 4 °C. The DNA was stained with ethidium bromide and visualized under UV light. 

### 4.9. Regulatory Experiments with Promoter-xylE Fusions in E. coli

The *E. coli* DH5α double transformants carrying pPT01 derivatives with the promoter regions of selected *P. aeruginosa* genes fused to the *xylE* reporter gene plus pAMB9.37 (*tac*p) or pKKB3.11 (*tac*p-*PA3973*) were assayed for catechol 2,3-oxygenase activity (the activity of the product of *xylE*). The experiment was performed as previously described [70] using cell extracts prepared from exponentially growing cultures in LB. Measurements of the amount of the protein in extracts were conducted using the Bradford assay [71].

## Figures and Tables

**Figure 1 ijms-23-14584-f001:**
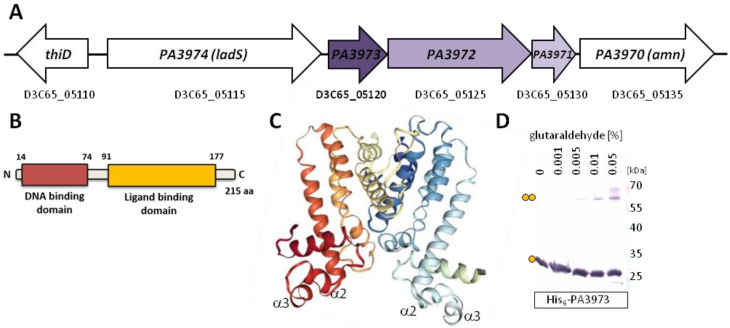
Properties of PA3973 protein from *P. aeruginosa*. (**A**) Genomic context of the *PA3973* gene in the *P. aeruginosa* PAO1 genome. The gene names from PAO1161 and PAO1 strains are presented below and inside the arrows representing loci, respectively. The *PA3973-PA3972-PA3971* gene cluster is shown in shades of purple. (**B**) Domain structure of PA3973 protein. (**C**) Structural model of PA3973 dimer obtained using AlphaFold2 and HDOCK [40,41]. Monomers are coloured in shades of red to khaki or navy blue to olive. The helices α2 and α3 constituting the HTH DNA binding domain are marked. (**D**) Analysis of the oligomerization state of purified His_6_-PA3973 by cross-linking with glutaraldehyde. Samples were separated by SDS-PAGE (12% gel) and analyzed by Western blot using anti-His_6_ antibodies. Dimeric forms are marked by two dots.

**Figure 2 ijms-23-14584-f002:**
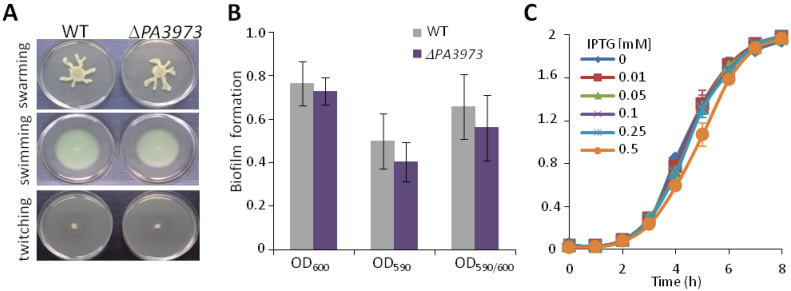
Phenotype analysis of *P. aeruginosa* PAO1161 WT, ∆*PA3973*, and PA3973 overproducer strains. (**A**) Selected pictures of swimming, swarming, and twitching analyses of WT *P. aeruginosa* PAO1161 and Δ*PA3973* mutant. (**B**) Biofilm formation of WT *P. aeruginosa* PAO1161 and Δ*PA3973* mutant. Cultures were grown without shaking in LB medium at 37 °C for 24 h and biofilm content measurements were performed. Data represent mean ±SD from 6 biological replicates. No statistically significant changes were observed by comparing Δ*PA3973* with WT (*p* > 0.05 in a two-sided Student *t* test). (**C**) Growth curves of *P. aeruginosa* PAO1161 cells carrying pKKB3.11 (*tac*p-*PA3973*) grown under selection in LB in the presence of different concentrations of IPTG. Data represent mean ± SD from 3 biological replicates.

**Figure 3 ijms-23-14584-f003:**
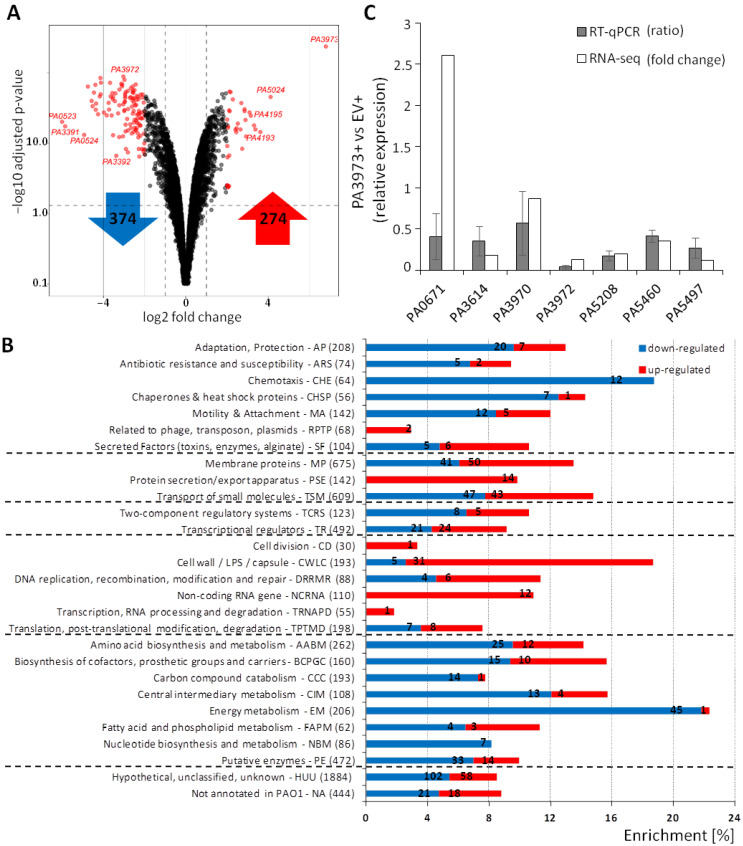
Identification of PA3973 dependent genes in *P. aeruginosa* using RNA-seq. Transcriptomes of PAO1161 cells carrying pKKB3.11 (*tac*p-*PA3973*) or pAMB9.37 (*tac*p), grown under selection in LB supplemented with 0.05 mM IPTG were analyzed by RNA-sequencing. (**A**) Volcano plot of RNA-seq data comparing the transcriptomes of PA3973+ and EV cells. Most significantly altered genes are indicated in red. The numbers of up- and down-regulated loci are presented in red and blue arrows, respectively. (**B**) Classification of loci (as in Appendix A) with altered expression in response to PA3973 excess according to PseudoCAP categories [11]. The bars indicate the enrichment of genes belonging to each category relative to all genes from the category in the PAO1 genome. Numbers presented in red and blue bars correspond to the numbers of up- and down-regulated genes, respectively, in each category. (**C**) Validation of RNA-seq data by RT-qPCR analysis. The RT-qPCR was performed using RNA samples obtained for the same conditions of growth as samples used for RNA-seq analysis. The results of RNA-seq data are presented as a fold change according to data presented in Appendix A, while RT-qPCR data represent the mean ratio ±SD for three biological replicates of PA3973+ cells relative to mean expression of the control EV+ strain.

**Figure 4 ijms-23-14584-f004:**
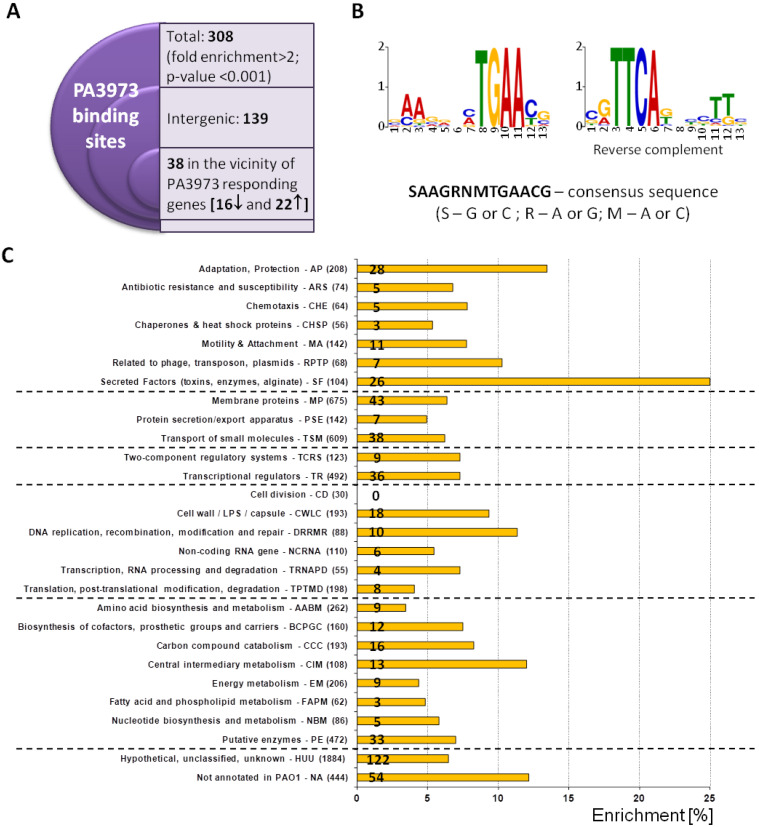
Identification of PA3973 binding sites in *P. aeruginosa* (ChIP-seq). (**A**) Cells expressing PA3973-FLAG were subjected to chromatin immunoprecipitation using anti-FLAG antibodies. Reads obtained by sequencing the ChIP DNA were mapped onto the PAO1161 genome. A gene was classified as likely to be directly regulated by PA3973 if the ChIP-seq peak summit was located in the vicinity of the start codon of the gene affected in RNA-seq by PA3973 excess. (**B**) Sequence logo of the PA3973 binding motif and its reverse complement version obtained by MEME tool [42]. The proposed consensus sequence is presented below. (**C**) Functional classification of genes likely to be affected by PA3973 binding. Bars present the enrichment of loci with PA3973 binding sites identified in ChIP-seq in the vicinity of genes and coding regions (identified genes from Appendix A), belonging to each PseudoCAP category [11]. Numbers indicate the exact number of genes belonging to each category with PA3973 binding site in their vicinity.

**Figure 5 ijms-23-14584-f005:**
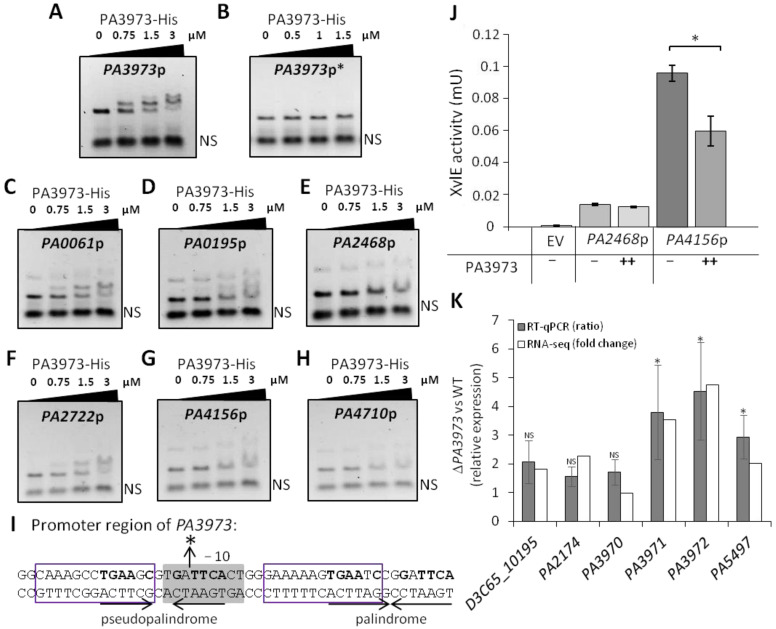
Regulatory properties of PA3973 assayed in vitro and in vivo. EMSA analysis of PA3973-His_6_ binding to regions preceding *PA3973* [long (**A**) and short promoter fragment (**B**)], *PA0061* (**C**), *PA0195* (**D**), *PA2468* (**E**), *PA2722* (**F**), *PA4156* (**G**), *PA4710* (**H**). Amplified DNA fragments were incubated with the indicated amounts of the protein in the presence of unspecific, competitor DNA (marked as NS), and complexes were separated by electrophoresis on 1.5% agarose gel. Ethidium bromide staining was used for the visualization of DNA. (**I**) Sequence preceding the *PA3973* gene. Probable binding motifs are marked with the purple box. “*” indicates the site of truncation of *PA3973*p to obtain its shorter fragment. (**J**) XylE activity in *E. coli* DH5α carrying pMEB267 (*PA2468*p-*xylE*) or pMEB269 (*PA4156*p-*xylE*) with pKKB3.11 (*tac*p-*PA3973*) (++) or empty plasmid pAMB9.37 (−). Strains were grown in selective LB with supplementation of 0.2 mM IPTG for pKKB3.11 (++). Cells carrying the promoter-less pPTOI (-*xylE*) (EV) and pAMB9.37 were used as background control. The data represent the means ± SD and * indicates *p* < 0.05 in a Student’s two-tailed *t*-test. (**K**) The genes with altered expression in Δ*PA3973* strain vs. WT PAO1161 as indicated by RNA-seq analysis and RT-qPCR analysis carried out using RNA samples obtained for the same conditions as samples used for RNA-seq analysis. The results of RNA-seq data are presented as a fold change according to data presented in Appendix A, while RT-qPCR data are presented as a ratio. Data represent mean ratio ± SD for three biological replicates. * indicate significantly different expression (ratio > 2, *p*-value < 0.05 in two-sided Student’s *t*-test assuming equal variance) comparing Δ*PA3973* with WT using the *proC* reference gene; NS—not significant.

**Figure 6 ijms-23-14584-f006:**
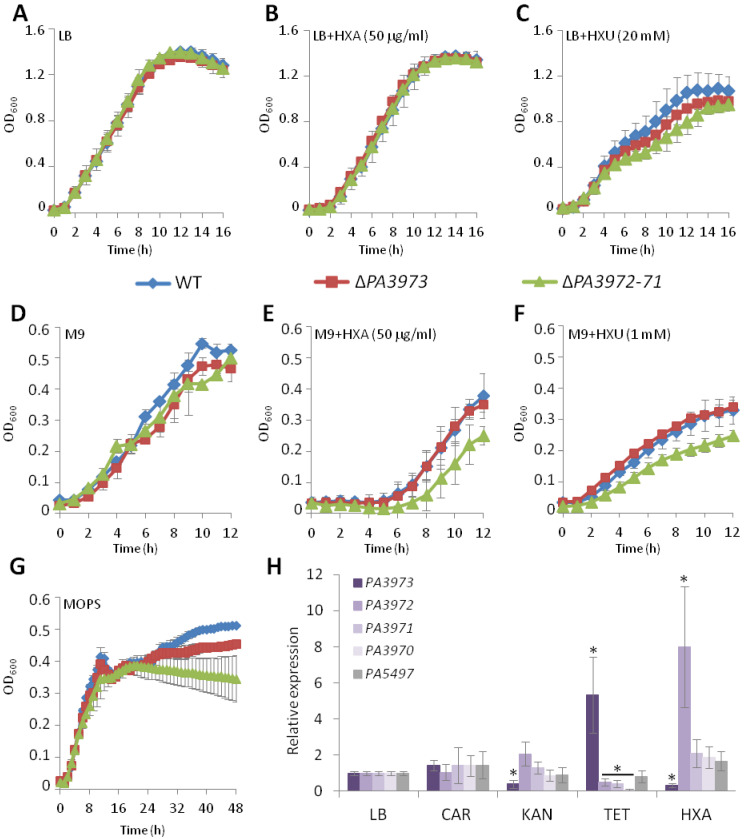
Involvement of *PA3973*-*PA3971* gene cluster in response of *P. aeruginosa* to stress. Kinetics of growth of *P. aeruginosa* strains PAO1161 WT, Δ*PA3973* and Δ*PA3972-71* at 37 °C in indicated media in the presence of hydroxylamine and hydroxyurea: LB (**A**), LB + 50 µg/mL hydroxylamine (**B**), LB + 20 mM hydroxyurea (**C**), M9 + glucose (**D**), M9 + glucose + 50 µg/mL hydroxylamine (**E**), M9 + glucose + 1 mM hydroxyurea (**F**), MOPS + glucose (**G**). Data represent mean ± SD from three independent experiments. HXA-hydroxylamine; HXU-hydroxyurea. (**H**) Relative expression of indicated genes in WT PAO1161 cells cultured in LB medium without antibiotic [LB] and with different classes of antibiotic added at subinhibitory concentrations: carbenicillin, 32 µg/mL [CAR]; kanamycin, 10 µg/mL [KAN], and tetracycline, 4 µg/mL [TET]. Additionally, the expression of indicated genes in cells grown with hydroxylamine (50 µg/mL) [HXA] added to the medium was tested and compared to the control [LB]. * *p* < 0.05 in a Student two-sided *t*-test assuming equal variance.

## Data Availability

Sequencing data are available in the NCBI’s Gene Expression Omnibus (GEO) database (http://www.ncbi.nlm.nih.gov/geo/) under accession number GSE211769 [ChIP-seq data] and GSE211771 [RNA-seq data] (released 22 November 2022).

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
