# Peer review of "Functional Characterization of TetR-like Transcriptional Regulator PA3973 from Pseudomonas aeruginosa"

_ijms, 2022, doi:10.3390/ijms232314584_

Round 1

Reviewer 1 Report

Major comments:

All the figures presented in the manuscript need to be replaced with higher-resolution ones, at least 300 ppi.

F1c: needs to clarify color representations in its figure legend
F1d: should include other members of the TetR family protein in the crosslinking assay as control. Also how to explain the doublet bands that are supposed to be “target dimers”?

F2: Since the authors managed to conduct RNA-seq analysis on the  KO strain(I assume it’s ΔPA3973), I would suggest the authors compile the sequencing data analysis with the WT and overexpressed strain(PAO1161), in order to systematically profile the regulon of PA3973.

F3: Parts of 3B and 3C cannot be seen in the manuscript. Please correct them.

F4:  It would be great to attach a list(as supplementary data) detailing the 38 genes overlapped between ChIP-seq and RNA-seq dataset.

F5: EMSA: Panel B: why not using the same concentrations of PA3973-His(eg, 0.75, 1,5 and 3uM) when testing the mutated/truncated promoter of pA3973?

Besides, given PA3973 belongs to TetR-type family whose members share high structural similarities, can the author predict the mutation sites (for example based on AlphaFold structure displayed in Fig 1C), which potentially would abrogate its DNA-binding affinity? It would be even better to include that mutated protein in EMSA setting to corroborate the findings.

F5K: I am not sure I fully understand this data. Can the authors elaborate on the ratio calculation/fold change as y-axis? What about any gene that is significantly downregulated upon knocking out PA3973? Also, where are the reference genes in this qPCR dataset?

F6: In terms of unveiling the biological function of this regulon, have the authors checked any of the regulon candidates downstream of PA3973? Maybe one can infer from the essentiality of regulon genes, therefore helping to address the importance of this transcriptional pathway.

Author Response

Reviewer 1

Thank you for all valuable suggestions and comments.

Comments and Suggestions for Authors, Major comments

All the figures presented in the manuscript need to be replaced with higher-resolution ones, at least 300 ppi. - Corrected.

F1c: needs to clarify color representations in its figure legend - We added description to the figure legend. Additionally we marked the positions of helices α2 and α3 constituting the HTH DNA binding domain of the protein.

“Monomers are coloured in shades of red to khaki or navy blue to olive. The helices α2 and α3 constituting the HTH DNA binding domain are marked.”

F1d: should include other members of the TetR family protein in the crosslinking assay as control. Also how to explain the doublet bands that are supposed to be “target dimers”?  -

- We repeated the crosslinking assay for His6-PA3973 including also in analysis the purified MexZ-His6, which is a member of TetR family of transcriptional regulators [Matsuo et al., 2004; Kawalek et al., 2019]. We confirmed that both proteins are able to form higher order complexes, preferentially dimers with different intensities. The target dimer in the case of MexZ is even harder to estimate in comparison with PA3973, as no clear single band, but rather smear of bands could be visible which may correspond to MexZ dimmers, possibly in different conformational states or after modification by an unknown factor. The gel demonstrating dimerization properties of tested proteins is shown below (Figure R1 and is also included to the document including original images attached to the submitted manuscript). The part demonstrating dimerization assay of His6-PA3973 is presented in the revised version of Figure 1D.

The dimerization assay for His6-PA3973, using a glutaraldehyde cross-linking agent repeatedly showed doublet bands visible above the target dimer. We hypothesize that they may reflect the different conformational states of the PA3973 dimer after crosslinking.

 (see attached pdf)                      

Figure R1. Analysis of the oligomerization state of purified His6-PA3973 and MexZ-His6 by cross-linking with glutaraldehyde. Samples were separated by SDS-PAGE (12% gel) and analyzed by Western blot using anti-His6 antibodies. Dimeric forms of PA3973 are marked by two dots.

F2: Since the authors managed to conduct RNA-seq analysis on the  KO strain(I assume it’s ΔPA3973), I would suggest the authors compile the sequencing data analysis with the WT and overexpressed strain(PAO1161), in order to systematically profile the regulon of PA3973.

- We added the RNA-seq data comparing PA3973+ vs EV+ to Table S6 for the list of genes showing significant changes in expression comparing DPA3973 vs WT. Importantly, most of the genes showed reverse regulation (increased expression in the absence of PA3973, and down-regulation upon PA3973 overproduction) in tested conditions, confirming the role of PA3973 as a repressor of these genes.

F3: Parts of 3B and 3C cannot be seen in the manuscript. Please correct them. - We apologize for that. The layout and positioning of figures was double-checked before re-submission.

F4:  It would be great to attach a list(as supplementary data) detailing the 38 genes overlapped between ChIP-seq and RNA-seq dataset. - These data can be found in Appendix A- Table A2 and should be visible in the manuscript file before the References section. We decided to attach this list in Appendix as it should be easier accessible for readers, than the supplement materials.

F5: EMSA: Panel B: why not using the same concentrations of PA3973-His(eg, 0.75, 1,5 and 3uM) when testing the mutated/truncated promoter of pA3973?

Besides, given PA3973 belongs to TetR-type family whose members share high structural similarities, can the author predict the mutation sites (for example based on AlphaFold structure displayed in Fig 1C), which potentially would abrogate its DNA-binding affinity? It would be even better to include that mutated protein in EMSA setting to corroborate the findings.

- We use the same concentrations of PA3973-His6 in EMSA assay testing different DNA fragments in the same experiment. In Figure 5B we presented representative data concerning truncated version of PA3973p from the gel when lower concentrations of the protein were tested (data for WT and truncated PA3973p version are showed for comparison below in Figure R2).

For the experiment in which higher concentrations of the protein were used, the fragment with mutated PA3973p was slightly less visible, but the results were the same, confirming the lack of interaction of PA3973-His6 with such mutated fragment.

(see attached pdf)    

Figure R2. EMSA analysis of PA3973-His6 binding to regions preceding PA3973 [* indicate truncated promoter fragment]. Amplified DNA fragments were incubated with the indicated amounts of the protein in the presence of unspecific, competitor DNA (the lowest band)), and complexes were separated by electrophoresis on 1.5% agarose gel. Ethidium bromide staining was used for the visualization of DNA.

We agree that including a mutant, with a point mutation blocking the binding to DNA (e.g. in HTH motif), would be the best control for the EMSA assay. In the current project we have focused on the sequence motif recognized by PA3973 and we have used a promoter variant lacking the potential site to prove this, which at the same time serves as specificity control for our EMSA assays. Nevertheless, analysis at molecular level is planned in the future.

F5K: I am not sure I fully understand this data. Can the authors elaborate on the ratio calculation/fold change as y-axis? What about any gene that is significantly downregulated upon knocking out PA3973? Also, where are the reference genes in this qPCR dataset? - Yes, it should be clarified and corrected. The results of RNA-seq data are presented as a fold change according to data presented in Table S6, while RT-qPCR data are presented as a ratio. The description was corrected in Figure 5K.

In RT-qPCR analysis the changes in individual gene expression (the relative expression ratio) between the WT and mutant strain were calculated with normalization of Cp values to mean Cp value for proC (PA0393) reference housekeeping gene using the Pfaffl method [Pfaffl, 2001].

According to second part of the comments there was no such significantly downregulated gene, as all genes with significantly changed expression in DPA3973 mutants cells showed increased mRNA level in comparison with WT cells as summarised in Table S6.

F6: In terms of unveiling the biological function of this regulon, have the authors checked any of the regulon candidates downstream of PA3973? Maybe one can infer from the essentiality of regulon genes, therefore helping to address the importance of this transcriptional pathway.

We focused on the analysis of PA3972 /PA3971 genes to unravel the biological function of their products, as these genes were directly regulated by PA3973. Further analyses are planned in the future to unveiling the biological function of other genes of PA3973 regulon.

References:

Kawalek, A.; Modrzejewska, M.; Zieniuk, B.; Bartosik, A.A.; Jagura-Burdzy, G. Interaction of ArmZ with the DNA-binding domain of MexZ induces expression of mexXY multidrug efflux pump genes and antimicrobial resistance in Pseudomonas aeruginosa. Antimicrob Agents Chemother 2019, AAC.01199-19, doi:10.1128/AAC.01199-19.

Matsuo, Y.; Eda, S.; Gotoh, N.; Yoshihara, E.; Nakae, T. MexZ-mediated regulation of MexXY multidrug efflux pump expression in Pseudomonas aeruginosa by binding on the mexZ-mexX intergenic DNA. FEMS Microbiol Lett 2004, 238, 23–28, doi:10.1016/j.femsle.2004.07.010.

Pfaffl, M.W. A New Mathematical Model for relative quantification in real-time RT-PCR. Nucleic Acids Res 2001, 29, e45, doi:10.1093/nar/29.9.e45.

Reviewer 2 Report

Review on

“Functional characterization of TetR-like transcriptional regulator PA3973 from Pseudomonas aeruginosa”

for IJMS (manuscript ID ijms-2029355)

The comprehensive introduction contains the historical perspective about mechanisms of antibiotics resistance of P. aeruginosa. The aim of this study is to uncover the functional role of protein PA3973 (TetR-type putative transcriptional regulator). This manuscript is relevant to the journal and its section “Molecular Microbiology”.

My questions about Results and Discussion:

1.     I cannot find the tables A1, A2 because no Appendix available in manuscript or supplementary. Might be better to include part of these tables into the manuscript.

2.     The confidence interval for D3C65_10195, PA3971, PA3972 includes the control values (Figure 5K). It shows that statistically significant difference is questionable and requires accurate statistical evaluation of the data. The same problem with the Figure 2B.

3.     Authors do not use the existing knowledge about PA3973 from PseudomonasNet, f.e. terms “negative regulation of transcription, DNA-dependent(EC)” and “regulation of transcription, DNA-dependent(PA)”. I would be better to discuss these pathways in introduction.

Methods section. GEO data are unavailable for accession numbers provided “Accession … is currently private and is scheduled to be released on Dec 31, 2022.”:

https://www.ncbi.nlm.nih.gov/geo/query/acc.cgi?acc=GSE211771

https://www.ncbi.nlm.nih.gov/geo/query/acc.cgi?acc=GSE211769

There are some improvements should be made with figures:

·        Figure 3 is partially invisible because of page margin.

·        What the meaning of the colors of Figure 1A?

Some minor corrections to the text (style and spelling):

·        File Table S3 contains caption of Figure 3B

·        No units at vertical axis at Figure 3C

·        L460-462: Might be better to place the explanation “ferric uptake regulation protein” to the first occurrence of “Fur”. The same case for “ribonucleotide reductases” at L472-473

Author Response

Responses to the Reviewers

Reviewer 2:

Thank you for all valuable suggestions and comments.

My questions about Results and Discussion:

  1. I cannot find the tables A1, A2 because no Appendix available in manuscript or supplementary. Might be better to include part of these tables into the manuscript.

We included both tables as Appendix A and they should be visible together with the manuscript before References section (p. 18-22). There could be a problem with conversion to pdf possibly (independent from us) which skips these section.

  1. The confidence interval for D3C65_10195, PA3971, PA3972 includes the control values (Figure 5K). It shows that statistically significant difference is questionable and requires accurate statistical evaluation of the data. The same problem with the Figure 2B. - Corrected and described according to performed statistic tests. For results presented in Figure 2B no statistically significant changes concerning biofilm formation were observed by comparing DPA3973 with WT (P > 0.05 in a two-sided Student t test).

For results presented in Figure 5K, according to RT-PCR data, the ratio higher than 2 comparing DPA3973 with WT indicate significantly different expression of gene (the ratio > 2, p-value <0.05 in two-sided Student's t-test assuming equal variance). Changes in individual gene expression between the WT and mutant strain were calculated with normalization of Cp values to mean Cp value for proC (PA0393) reference housekeeping gene using the Pfaffl method [Pfaffl, 2001].

  1. Authors do not use the existing knowledge about PA3973 from PseudomonasNet, f.e. terms “negative regulation of transcription, DNA-dependent(EC)” and “regulation of transcription, DNA-dependent(PA)”. I would be better to discuss these pathways in introduction. - We added the description in introduction (p. 3). “Based on the PseudomonasNet, a genome-wide functional network of P. aeruginosa genes, a negative regulation of transcription, DNA-dependent and cellular response to stress are predicted for PA3973 according to GO terms [Hwang et al., 2016].”

Methods section. GEO data are unavailable for accession numbers provided “Accession … is currently private and is scheduled to be released on Dec 31, 2022.”:

https://www.ncbi.nlm.nih.gov/geo/query/acc.cgi?acc=GSE211771

https://www.ncbi.nlm.nih.gov/geo/query/acc.cgi?acc=GSE211769

- We apologize for this inconvenience. We have provided the secure tokens allowing review of GSE211771 and GSE211769 records while they remain in private status, but in the cover letter, probably not available for Reviewers.

The following secure tokens have been created to allow review of GSE211771 and GSE211769 records while they remain in private status.

To review GEO accession GSE211769:

Go to https://www.ncbi.nlm.nih.gov/geo/query/acc.cgi?acc=GSE211769

Enter token uhmtyguuxpazhmd into the box

To review GEO accession GSE211771:

Go to https://www.ncbi.nlm.nih.gov/geo/query/acc.cgi?acc=GSE211771

Enter token gjmxqyymbnipdof into the box.

The data supporting the results of this article were deposited in the NCBI‘s Gene Expression Omnibus (GEO) database and will be accessible through GEO Series accession number [GSE211771 – RNA-seq data and GSE211769 – ChIP-seq data] after publication acceptance.

There are some improvements should be made with figures:

  • Figure 3 is partially invisible because of page margin. - Corrected.
  • What the meaning of the colors of Figure 1A? - We added description to the figure legend. “The PA3973-PA3972-PA3971 gene cluster is shown in shades of purple.”

Some minor corrections to the text (style and spelling):

  • File Table S3 contains caption of Figure 3B – Corrected.
  • No units at vertical axis at Figure 3C - The description of the figure was corrected. The relative expression ratio of genes with altered expression comparing PA3973+ vs EV was presented.
  • L460-462: Might be better to place the explanation “ferric uptake regulation protein” to the first occurrence of “Fur”. The same case for “ribonucleotide reductases” at L472-473 – Corrected.

References:

Hwang, S.; Kim, C.Y.; Ji, S.-G.; Go, J.; Kim, H.; Yang, S.; Kim, H.J.; Cho, A.; Yoon, S.S.; Lee, I. Network-assisted investigation of virulence and antibiotic-resistance systems in Pseudomonas aeruginosa. Sci Rep 2016, 6, 26223, doi:10.1038/srep26223.

Pfaffl, M.W. A New Mathematical Model for relative quantification in real-time RT-PCR. Nucleic Acids Res 2001, 29, e45, doi:10.1093/nar/29.9.e45.
